# Discovery of Three New *Mucor* Species Associated with Cricket Insects in Korea

**DOI:** 10.3390/jof8060601

**Published:** 2022-06-03

**Authors:** Thuong T. T. Nguyen, Hyang Burm Lee

**Affiliations:** Environmental Microbiology Lab, Department of Agricultural Biological Chemistry, College of Agriculture & Life Sciences, Chonnam National University, Gwangju 61186, Korea; ngthuongthuong@gmail.com

**Keywords:** cricket insect, ITS, LSU, *Mucorales*, phylogeny, taxonomy

## Abstract

Species in the genus *Mucor* have a worldwide distribution and are isolated from various substrata and hosts, including soil, dung, freshwater, and fruits. However, their diversity from insects is still much too little explored. The aim of this study was to characterize three new species of *Mucor*: *Mucor grylli* sp. nov., *M*. *hyangburmii* sp. nov., and *M*. *kunryangriensis* sp. nov., discovered in Kunryang-ri, Cheongyang in the Chungnam Province of Korea, during an investigation of *Mucorales* from cricket insects. The new species are described using morphological characters and molecular data including ITS and LSU rDNA regions. *Mucor grylli* is characterized by the highly variable shape of its columellae, which are subglobose to oblong, obovoid, strawberry-shaped, and sometimes slightly or strongly constricted in the center. *Mucor hyangburmii* is characterized by the production of azygospores and growth at 40 °C. *Mucor kunryangriensis* is characterized by the variable shape of its columellae, which are elongated-conical, obovoid, cylindrical ellipsoid, cylindrical, and production of abundant yeast-like cells on PDA, MEA, and SMA media. Based on the sequence analysis of two genetic markers, our phylogenic assessment strongly supported *M. grylli*, *M*. *hyangburmii,* and *M*. *kunryangriensis* as new species. Detailed descriptions, illustrations, and phylogenetic trees are provided.

## 1. Introduction

The genus *Mucor* was described by Fresenius [1], and it is classified in the family *Mucoraceae*, order *Mucorales*, and phylum *Mucoromycota*, which belongs to the early diverging fungi [2]. With more than 90 currently accepted species [3,4,5,6,7,8], *Mucor* is the largest genus within the *Mucorales*. Species of *Mucor* are known to be saprotrophs that are usually isolated from dung, soil, freshwater, insects, or fruits [6,9,10,11,12,13,14]. Some species are human pathogens causing mucormycosis [15], and to date, 12 species are known to be involved in infections [3,16]. *Mucor* species have important industrial applications because of their ability to produce a wide range of metabolites [17,18]. Some *Mucor* species produce enzymes such as protease, phytase, cellulase, lipase, and uricase [18,19,20,21,22,23]. Moreover, some species are used to manufacture Asian fermented food products and beverages [24,25]. The study published by Walther et al. [10], which included more than 300 *Mucor* strains, placed species of *Mucor* in different groups. Those groups were intermingled in the LSU phylogenetic tree with species of other genera. The results were that species of *Mucor* divide into six groups, consisting of the *M*. *mucedo* group, the *M*. *flavus* group, the *M*. *hiemalis* group, the *M*. *racemosus* group, the *M*. *amphibiorum* group, and the *M*. *recurvus* group [10]. The *M. amphibiorum* group contains two species that are potentially involved in human infections: *M*. *amphibiorum* and *M*. *ardhlaengiktus* [3]. In recent years, the number of new species of *Mucor* has increased. However, there were only four species recorded in the *M*. *amphibiorum* group from 2015 to February 2022, including *M. caatinguensis* A.L. Santiago, C.A.F. de Souza & D.X. Lima [26], *M*. *fluvii* Hyang B. Lee, S.H. Lee & T.T.T. Nguyen [11], *M*. *pernambucoensis* C.L. Lima, D.X. Lima & A.L. Santiago [27], and *M*. *chiangraiensis* V.G. Hurdeal, E. Gentekaki, K.D. Hyde & H.B. Lee [5]. Most of those species were isolated from soil [5,26,27], except for *M*. *fluvii*, which was isolated from freshwater [11]. 

The purpose of this study was to expand the present knowledge of the fungal diversity found in poorly studied substrates or unexplored areas. Herein, we describe and illustrate three new species of *Mucor* isolated from insects in Korea. 

## 2. Materials and Methods

### 2.1. Sampling and Isolation 

Cricket insect (*Gryllus* sp.) samples were collected from Kunryang-ri, Cheongyang, Chungnam Province, Korea, between April 2020 and October 2021. The insects were collected in polyethylene bags, stored at ambient temperature, and transported to the laboratory. Fungal isolation from the insect samples was conducted following our previous methods [28]. Briefly, the samples were transferred to clean Petri dishes. The insect bodies were then broken up into small pieces and placed on PDA. The plates were then incubated at 25 °C for 2–5 days. Then, hyphal tips were transferred to fresh PDA. All isolates were purified by single spore isolation as previously described [28]. 

Ex-type living cultures were deposited at Environmental Microbiology Laboratory Fungarium, Chonnam National University (CNUFC), Gwangju, Korea and the Culture Collection of National Institute of Biological Resources (NIBR), Incheon, Korea. Dried cultures were deposited in the Herbarium Chonnam National University, Gwangju, Korea.

### 2.2. Morphological Studies 

Pure cultures of *Mucor* spp. were cultured on potato dextrose agar (PDA), malt extract agar (MEA: 40 g malt extract, 4 g yeast extract, and 15 g agar in 1 L deionized water), and synthetic mucor agar (SMA: 40 g dextrose, 2 g asparagine, 0.5 g KH_2_PO_4_, 0.25 g MgSO_4_·7H_2_O, 0.5 mg thiamine hydrochloride, and 15 g agar in 1 L deionized water) [4,9]. The plates were incubated at 25, 30, 37, 39, 40, and 44 °C in the dark for 7 days. Fragments of mycelia were removed from the cultures and placed onto microscopy slides with 60% lactic acid. An Olympus BX53 microscope (Olympus, Tokyo, Japan) possessing differential interference contrast optics was used to obtain digital images. 

### 2.3. DNA Extraction, PCR, Cloning, and Sequencing

Total genomic DNA was extracted from fresh fungal mycelia that were grown on cellophane at 25 °C after 4 days using the Solg^TM^ Genomic DNA Preparation Kit (Solgent Co. Ltd., Daejeon, Korea) according to the manufacture’s protocol, and then stored at −20 °C. Two regions were amplified, including the internal transcribed spacer (ITS) region using primers ITS1 and ITS4 [29], and the large subunit rDNA region using primers LR0R and LR5 [30]. The PCR products were purified with the Accuprep PCR Purification Kit (Bioneer Corp., Daejeon, Korea) and sequenced at Macrogen (Daejeon, South Korea). Direct sequencing of the ITS PCR product failed; thus, we performed the cloning. PCR products after gel purification were ligated into the pGEM-T Easy Vector (Promega, Madison, WI, USA), following the manufacturer’s instructions. The ligation mixture was transformed into *Escherichia coli* DH5α by heat shock. The positive white colonies were grown in Luria broth (LB) media containing 100 μg of ampicillin per milliliter. The plasmids were purified using the Plasmid Purification Mini Kit (Nucleogen, Si-heung, South Korea). Then, purified plasmids of three clones were sequenced using the primers M13F forward (5′-GTAAAACGACGGCCAGT-3′) and M13R reverse (5′-GCGGATAACAATTTCACACAGG-3′).

### 2.4. Phylogenetic Analyses 

DNA sequences were checked and were assembled by Seqman Pro 7.1.0 in Lasergene package (DNASTAR, Madison, WI, USA). All newly generated sequences were submitted to GenBank database under the accession numbers provided in Table 1.

Sequence data of closely related *Mucor* spp. were selected from data previously published by Walther et al. [10], Li et al. [26], Wanasinghe et al. [11], Lima et al. [27], and Hurdeal et al. [5], and downloaded from GenBank (https://www.ncbi.nlm.nih.gov/genbank/ (accessed on 10 March 2022) [31] for molecular phylogenetic analyses (Table 1). Sequences of datasets ITS (44 taxa) and LSU (45 taxa) were aligned using MAFFT (http://mafft.cbrc.jp/alignment/server (accessed on 12 March 2022) with the algorithm L-INS-I [32], and the alignment was checked in MEGA7 [33]. Aligned sequences were automatically trimmed using trimAl with the gappyout method [34]. Data were converted from fasta format to nexus and phylip formats using the online tool Alignment Transformation Environment (https://sing.ei.uvigo.es/ALTER/ (accessed on 12 March 2022). Phylogenetic reconstructions by maximum likelihood (ML) and Bayesian inference (BI) were carried out using PhyML 3.0 [35], and MrBayes 3.2.2 [36], respectively. The most appropriate model was obtained using the software jModelTest v.2.1.10 [37,38]. We performed the ML analysis using 1000 bootstrap replicates under the best substitution model for the ITS (TPM2uf+I+G) and LSU (TIM3+I+G). BI analyses were performed using 5 million Markov chain Monte Carlo (MCMC) generations and the best substitution model HKY+G and HKY+I+G for ITS and LSU, respectively. The sample frequency was set to 100, the first 25% of trees were removed as burn-in, and the remaining trees were used to calculate the posterior probabilities. The resulting trees were viewed using FigTree v.1.3.1 (http://tree.bio.ed.ac.uk/software/figtree/ (accessed on 12 March 2022). The alignment files generated for phylogenetic analyses are provided in the Appendix A.

## 3. Results

### 3.1. Molecular Phylogenetic Analysis 

To understand the evolutionary relationship between isolated strains, sequences of ITS and LSU were used for the phylogenetic analysis (Figure 1 and Figure 2). The ITS and LSU phylogenetic analyses revealed that isolate CNUFC CY22 grouped with *M. pernambucoensis*, having strong statistical support in the ITS tree (ML/BI = 99/1) (Figure 1). However, this relationship was supported with a moderate bootstrap percentage (86%), but a low posterior probability value (<0.90) in the LSU tree (Figure 2). Isolate CNUFC CY223 was closely related to *M*. *zachae* in the ITS tree with high statistical support (ML/BI = 99/0.99) (Figure 1) but formed an independent clade with high statistical support (ML/BI = 99/0.99) in the LSU tree (Figure 2). In the ITS analysis (Figure 1), CNUFC CY102 was sister clade to *M*. *ucrainicus* with high statistical support (ML/BI = 94/0.92) and clustered with *M*. *odoratus* in our LSU (Figure 2) with a moderate posterior probability value (0.81) and a low bootstrap percentage (< 70%).

### 3.2. Taxonomy

Based on our phylogenies and morphological data, three new species of *Mucor* from cricket insects in Korea were described and illustrated here. 

***Mucor grylli*** Hyang B. Lee & T.T.T. Nguyen sp. nov. (Figure 3).

Index Fungorum: 555247.

Etymology: Referring to the host, *Gryllus* sp., from which the species was first isolated.

Type: REPUBLIC OF KOREA: Kunryang-ri (36°26′16.2” N 126°46′04.6” E), Cheongyang-eup, Cheongyang, Chungnam Province, from *Gryllus* sp., 20th June 2021, J.S. Kim; holotype CNUFC HT2102; ex-type living culture, CNUFC CY102). 

Description: Colonies on MEA at 25 °C at first white, becoming light gray, reaching 70–73 mm in diameter after 4 days of incubation; reverse uncolored. Sporangiophores erect, 6–14.5 µm diam., slightly sympodially branched with long branches. Sporangia globose, wall echinulate, (28–) 42–73.5 µm diam., slightly yellow, deliquescent in mature sporangia, and persistent in young sporangia. Columellae highly variable in shape, oblong (57.5–42.0 × 40–31.5 µm), subglobose, obovoid, or strawberry-shaped (25–44 × 18.5–35.5 µm), sometimes slightly or strongly constricted in the center (27.9–42.5 × 18.5–26.5 µm), with distinct collar. Sporangiospores mostly ellipsoid, 8.0–10.5 × 3–4 µm, usually with granules at each end. Chlamydospores formed on hyphae, terminal and intercalary, single, smooth and thick-walled, ellipsoidal. Zygospores not observed. On SMA and PDA, sporangia are larger (SMA: 42.5–105 µm diam.; PDA: 55.5–112 µm diam.) than on MEA. Columellae on SMA (up to 75 × 65.5 µm) are larger than PDA and MEA. Sporangiospores on SMA, MEA, and PDA are similar. 

Culture characteristics: On PDA, the colonies attain a diameter of 61–63 mm after 4 days at 25 °C. On SMA, the colonies attain a diameter of 66–69 mm after 4 days at 25 °C. At 37 °C on SMA, PDA, and MEA, growth is observed but sporulation is lacking. The colony reaches a diameter of 19, 22, and 18 mm at 37 °C after 4 days on PDA, SMA, and MEA, respectively. No growth was observed at 39 °C. 

**Figure 3 jof-08-00601-f003:**
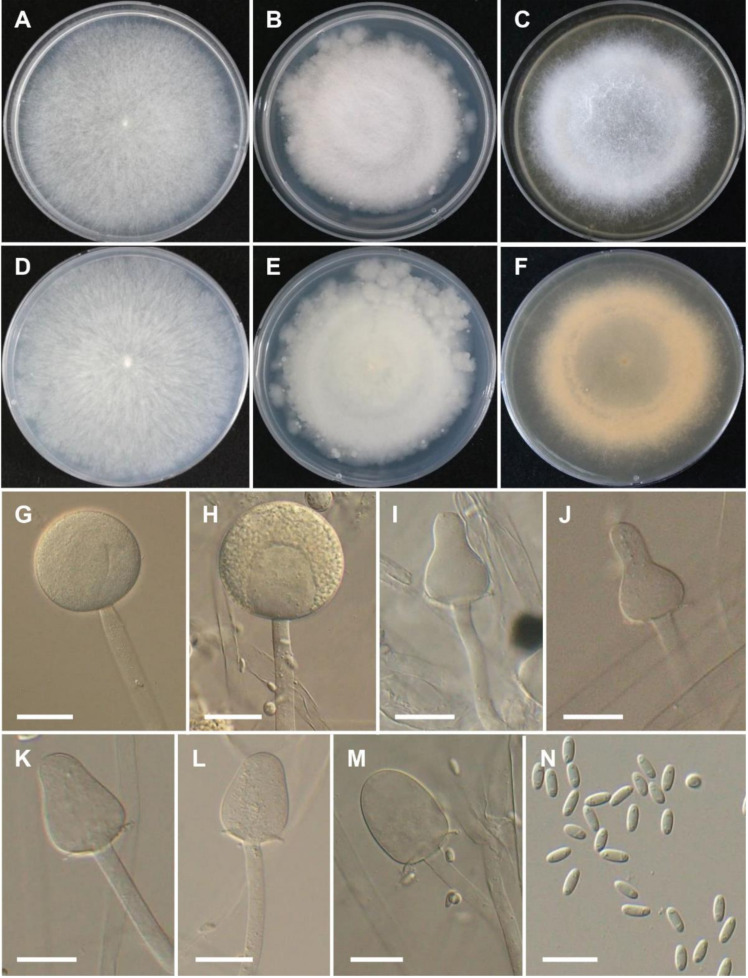
Morphology of *Mucor grylli*. (**A**,**D**) Colony on synthetic agar mucor (SMA). (**B**,**E**) Colony on potato dextrose agar (PDA). (**C**,**F**) Colony on malt extract agar (MEA). (**G**,**H**) Young and mature sporangia. (**I**–**M**) Typical columellae. (**N**) Sporangiospores. Scale bars = 20 µm.

***Mucor hyangburmii*** T.T.T Nguyen sp. nov. (Figure 4).

Index Fungorum: 555248.

Etymology: In honor of Dr. Hyang Burm Lee, a Korean mycologist who has studied basal fungal lineages in Korea and supervised the first author’s Ph.D. work.

Type: REPUBLIC OF KOREA: Kunryang-ri (36°26′16.2′′ N 126°46′04.6′′ E), Cheongyang-eup, Cheongyang, Chungnam Province, from the surface of leg of *Gryllus* sp., 28 July 2020, J.S. Kim; holotype CNUFC HT2021; ex-type living culture, CNUFC CY22.

Description: Colonies on MEA at 25 °C at first white, soon becoming pale, the central part with abundant azygospores, reaching 65–70 mm in diameter after 4 days of incubation; reverse moderate yellow. Sporangiophores erect, rare branched, 5–9 µm diam. The branches commonly bear a sterile sporangium that may form a new sporangium. Sporangia globose, yellow, (29–) 35.5–58.5 (–62) µm diam., rapidly deliquescent. Columellae globose, subglobose, (17–) 23.5–38.5 × (16–) 21.5–36 µm; collar present. Sporangiospores smooth, mostly ellipsoidal, (5.5–) 6.0–9.5 (–10.5) × (2.5–) 3.0–4.0 (–4.5) µm, sometime flattened at one side. Chlamydospores present in sporangiophores. Azygospores abundant, formed terminally on simple or branched azygophores, deep reddish brown subglobose, 19–45 µm diam. Zygospores were not observed. Colonies on SMA gray, chestnut in central part with abundant azygospores. Sporangiospores on SMA slightly smaller (5–8.5 × 2.5–4.0 µm) than on PDA and MEA. Chlamydospores are abundant and less sporangia on SMA media, but azygospores abundant and formed earlier than PDA and MEA. On PDA, sporangia slightly smaller (up to 56 µm diam.) than on MEA.

Culture characteristics: On PDA, the colonies attain a diameter of 55–57 mm after 4 days at 25 °C. On SMA, the colonies attain a diameter of 50–52 mm after 4 days at 25 °C. At 37 °C, colony reaches a diameter of 60, 67, and 61 mm after 4 days on PDA, SMA, and MEA, respectively. At 40 °C in SMA, PDA, and MEA, growth is observed but with no sporulation. The colony reaches a diameter of 31, 30, and 16 mm at 40 °C after 4 days on PDA, SMA, and MEA, respectively. No growth was observed at 44 °C. 

**Figure 4 jof-08-00601-f004:**
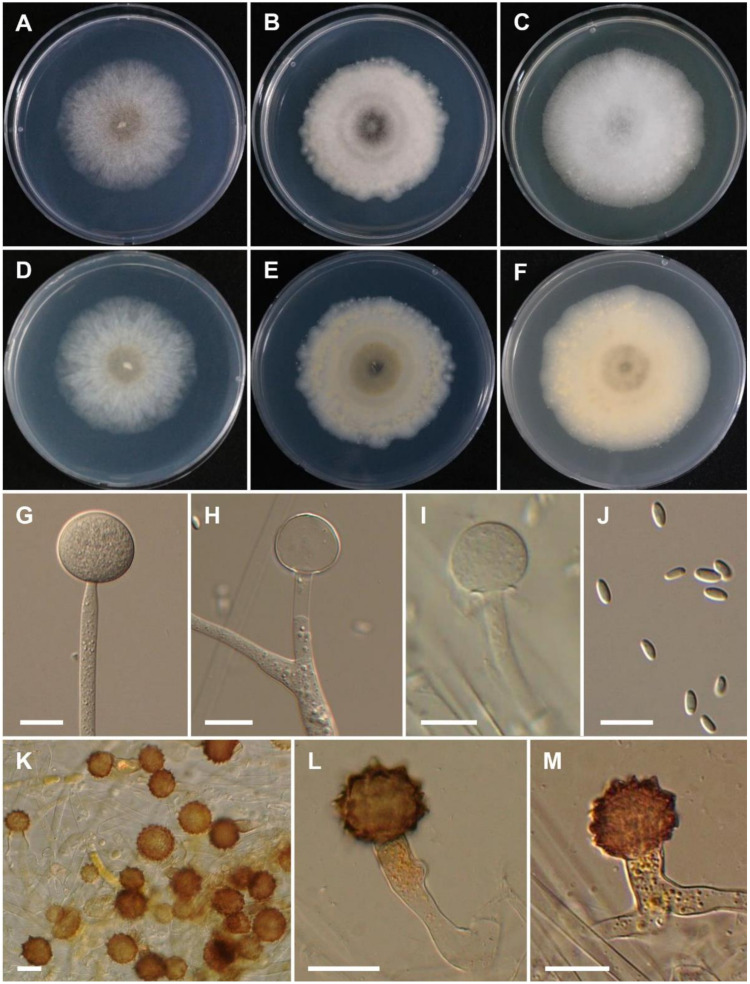
Morphology of *Mucor hyangburmii*. (**A**,**D**) Colony on synthetic agar mucor (SMA). (**B**,**E**) Colony on potato dextrose agar (PDA). (**C**,**F**) Colony on malt extract agar (MEA). (**G**) Young sporangium. (**H**) Sterile sporangium. (**I**) Columella. (**J**) Sporangiospores. (**K**–**M**) Azygospores on MEA. Scale bars = 20 µm.

***Mucor kunryangriensis*** Hyang B. Lee & T.T.T. Nguyen sp. nov. (Figure 5).

Index Fungorum: 555249.

Etymology: Referring to the isolation location, Kunryang-ri from where the species was first isolated (Korea).

Type: REPUBLIC OF KOREA: Kunryang-ri (36°26′16.2′′ N 126°46′04.6′′ E), Cheongyang-eup, Cheongyang, Chungnam Province, from *Gryllus* sp., 9th August 2021, H.B. Lee and J.S. Kim; holotype CNUFC HT2105; ex-type living culture, CNUFC CY223).

Description: Colonies on MEA at 25 °C white to grayish white, reaching 62–65 mm in diameter after 4 days of incubation; reverse light yellowish brown. Sporangiophores erect, unbranched or once branched, 4–7 (–10) µm diam. Sporangia yellow to light brown, globose, (20.5–) 24–45.5 (–47.5) µm diam. Columellae elongated-conical, obovoid, cylindrical ellipsoid, cylindrical, (13–) 18–26.5 × (8.5–) 10.5–14.5 μm. Sporangiospores mostly ellipsoid, sometimes flattened at one side, containing granules at each end, 5.0–7.5 × 2–3 μm. Yeast-like cells were abundant on PDA, SMA, and MEA, globose, 12.5–20.5 μm diam. Zygospores not observed. On SMA, sporangia are smaller (17.5–39 µm diam.) than on MEA and PDA [(18–) 23.5–44.5 µm diam.]. Sporangiospores on SMA and MEA are similar, but slightly larger on PDA (5.5–8.0 × 2–3.5 µm). 

Culture characteristics: On PDA, the colonies attain a diameter of 56–59 mm after 4 days at 25 °C. On SMA, the colonies attain a diameter of 29–32 mm after 4 days at 25 °C. At 37 °C on SMA, PDA, and MEA, growth is observed but with no sporulation. The colony reaches a diameter of 12, 13, and 21 mm at 37 °C after 4 days on PDA, SMA, and MEA, respectively. No growth was observed at 40 °C. 

**Figure 5 jof-08-00601-f005:**
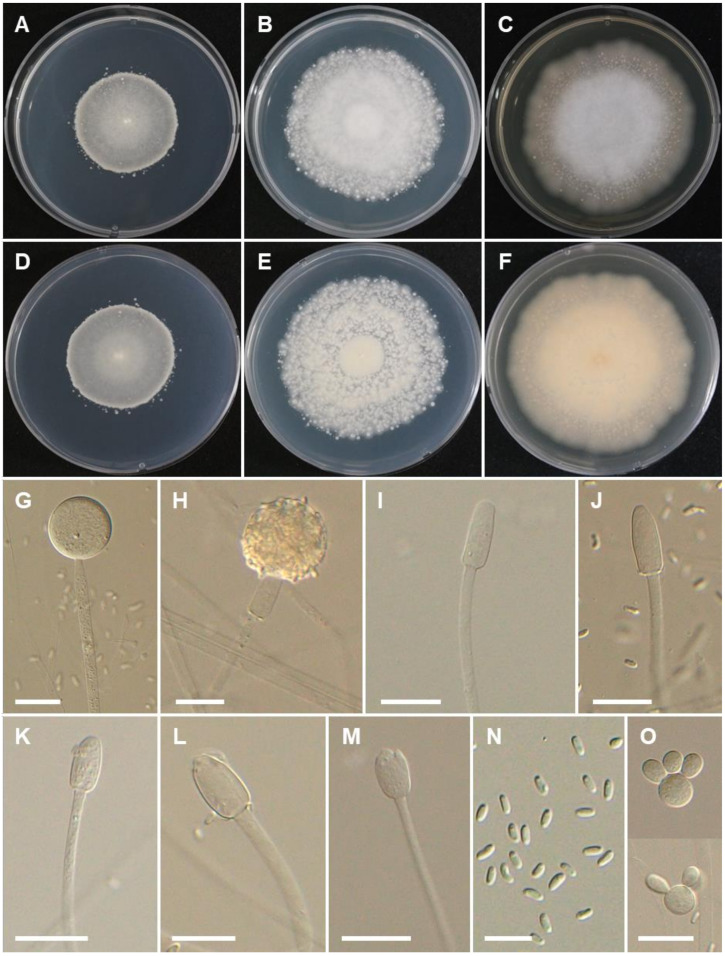
Morphology of *Mucor kunryangriensis*. (**A**,**D**) Colony on synthetic agar mucor (SMA). (**B**,**E**) Colony on potato dextrose agar (PDA). (**C**,**F**) Colony on malt extract agar (MEA). (**G**,**H**) Young and mature sporangia. (**I**–**M**) Typical columellae. (**N**) Sporangiospores. (**O**) Yeast-like cells. Scale bars = 20 µm.

## 4. Discussion

This study reports on new *Mucor* species isolated from *Gryllus* insects collected from Kunryang-ri, Cheongyang, located in Chungnam Province, Korea. Three new *Mucor* species belonging to the *M*. *amphibiorum* group [10] are described. Members of the *M*. *amphibiorum* group are characterized by unbranched tall sporangiophores or sporangia with a maximum diameter of between 70–175 µm [10]. 

Phylogenetic analyses of ITS and LSU showed that *M. grylli*, *M*. *ucrainicus*, *M*. *zychae,* and *M*. *odoratus* are phylogenetically close species. Based on Blastn searches, the ITS and LSU sequences of *M. grylli* were most similar to *M*. *zychae* (GenBank NR_103641; 477/558 bp (85.5%), *M*. *variisporus* (GenBank NR_152951; 435/512 bp (84.9%), *M*. *zychae* (GenBank NR_057930; 652/675 bp (96.6%), and *M*. *odoratus* (GenBank NR_057927; 659/690 bp (95.5%), respectively. However, *M*. *grylli* differs from these species by production of strawberry-shaped columellae, sometimes slightly or strongly constricted in the center. Wagner et al. [4] reported the presence of strawberry-shaped columellae in *M*. *variicolumellatus*, but *M*. *grylli* produces larger sporangiospores and includes granules at the end. Moreover, *M*. *grylli* can grow at 37 °C, while *M*. *variicolumellatus* cannot [4]. Treschew [39] has mentioned that *M*. *odoratus* grew slowly at 40 °C, whereas *M*. *grylli* was not able to grow at 40 °C. *Mucor ucrainicus* differs from *M*. *grylli* in producing larger sporangia (up to 175 µm diam.) and smaller sporangiospores (4.4–8.1 × 2.7–4.7 µm) [40].

*Mucor hyangburmii* is phylogenetically related to *M*. *pernambucoensis*, *M*. *ucrainicus,* and *M*. *variisporus* in the ITS and LSU trees (Figure 1 and Figure 2). Based on Blastn search, the ITS and LSU sequences of *M*. *hyangburmii* were most similar to *M*. *variisporus* (GenBank NR_152951; 334/379 bp (88.1%), *M*. *azygosporus* (GenBank NR_103639; 332/378 bp (87.8%), *M*. *ardhlaengiktus* (GenBank NR_069778; 661/693 bp (95.4%), and *M*. *azygosporus* (GenBank NR_057928; 661/693 bp (95.4%), respectively. However, the production of azygospores can easily distinguish *M*. *hyangburmii* from these species, except for *M*. *azygosporus*. Unlike *M*. *hyangburmii*, colony color on SMA of *M*. *azygosporus* is orange to buff orange [41]. In addition, *M*. *pernambucoensis*, *M*. *ucrainicus,* and *M*. *variisporus* displayed limited growth at 35, 30, and 37 °C [27,40,42], respectively, but *M*. *hyangburmii* was able to grow even at 40 °C. Sporangiospores of *M*. *variisporus* are larger and variable in shape and size (5.5–13.5 × 3.5–8 μm) [42] than those of *M*. *hyangburmii*, which are mostly ellipsoidal [(5.5–) 6.0–9.5(–10.5) × (2.5–) 3.0–4.0 (–4.5) µm]. *Mucor hyangburmii* shared some similarities with *M*. *pernambucoensis*, such as ellipsoidal sporangiospores [27]. However, sporangiospores of *M*. *pernambucoensis* reported by Lima et al. [27] were slightly larger (4.5–12 (–14.5) × 2.5–5 µm) than those of the isolate obtained in this study. In addition, columellae of *M*. *pernambucoensis* are globose, obovoid, cylindrical, and pyriform, differing from those of *M. hyangburmii* that are globose and subglobose. 

The phylogenetic analysis of ITS and LSU show that *M*. *kunryangriensis* is phylogenetically related to *M*. *zychae*, *M*. *odoratus*, and *Mycotypha microspora* (Figure 1 and Figure 2). Based on Blastn search, the ITS and LSU sequences of *M*. *kunryangriensis* were most similar to *M*. *zychae* (GenBank NR_103641; 485/539 bp (89.9%), *M*. *odoratus* (GenBank NR_145287; 520/590 bp (88.1%), *M*. *zychae* (GenBank NR_057930; 651/671 bp (97.1%), and *M*. *ardhlaengiktus* (GenBank NR_069778; 660/686 bp (96.2%), respectively. However, the new species can be easily distinguished from these species by the production of abundant yeast-like cells and columellae that are elongated-conical or cylindrical. In addition, sporangia and sporangiospores of *M*. *kunryangriensis* are smaller than those of *M*. *odoratus* [sporangia: up to 100 µm diam.; sporangiospores: (7.8–) 9.5–17.5 (–21.6) × (3.7–) 4–9.8 (–13.5) µm] [39]. Sporangia of *M*. *zychae* are larger (up to 70 µm diam.) [43] than those of *M*. *kunryangriensis*. As observed in *M*. *kunryangriensis*, *M*. *guilliermondii*, *M*. *chuxiongensis,* and *M. gigasporus* also produce elongated-conical and cylindrical columellae. However, *M*. *chuxiongensis* differs from *M*. *kunryangriensis* by the production of smaller columellae (12–15 × 5.0–8.5 µm) and sporangiospores (4.5–6.5 × 2.0–2.5 µm) [44]. *Mucor guilliermondii* and *M*. *gigasporus* differ from *M*. *kunryangriensis* by producing larger sporangia (up to 60 µm diam. for *M*. *guilliermondii* and (35.6–) 45.7–76.2 (–88.9) µm diam. for *M*. *gigasporus*) [45,46]. 

It is also noteworthy that *M*. *grylli*, *M*. *hyangburmii*, and *M*. *kunryangriensis* are the first species in the *M*. *amphibiorum* group collected on insects. Interestingly, the three new species can grow optimally at near-human-body temperature, which needs attention as a potential cause of diseases. 

*Mucor* species are dimorphic fungi and exhibit either hyphal or yeast growth depending upon the conditions such as cultivation time, temperature, presence or absence of oxygen, and carbon and nitrogen sources [47,48]. Several studies reveal that *M*. *indicus* in different morphologies (filamentous and yeast-like forms) can produce ethanol with relatively high yields and productivity [49,50]. Interestingly, *M*. *kunryangriensis* also produces a yeast-like form, necessitating further studies. 

Three new species described here were recovered from samples collected at Kunryang-ri, Cheongyang, located in Chungnam Province, Korea, an area recognized as a biodiversity hotspot and known as the “Alps of Chungnam” with significant mucoralean species richness [13,28]. With further investigations, we expect to discover additional unreported species in this genus. New species could be a source of novel drugs and other useful compounds.

## Figures and Tables

**Figure 1 jof-08-00601-f001:**
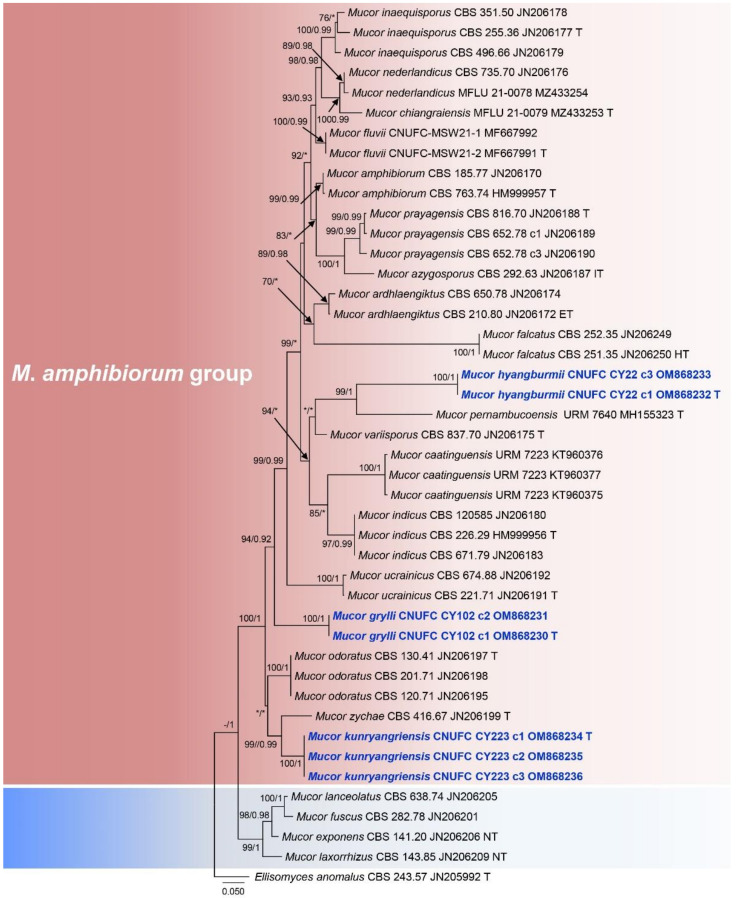
Phylogenetic tree constructed by maximum likelihood analysis of the ITS. The numbers above branches represent maximum likelihood bootstrap percentages (left) and Bayesian posterior probabilities (right). Bootstrap values ≥70% and Bayesian posterior probabilities ≥0.90 are shown. Bootstrap values lower than 0.90 and 70% are marked with “*”, and absent bootstrap values are marked with “-”. The bar indicates the number of substitutions per position. *Ellisomyces anomalus* CBS 243.57 was used as outgroup. Type, ex-neotype, ex-holotype, and ex-epitype strains are marked with T, NT, HT, and ET, respectively. Newly generated sequences are in blue bold font.

**Figure 2 jof-08-00601-f002:**
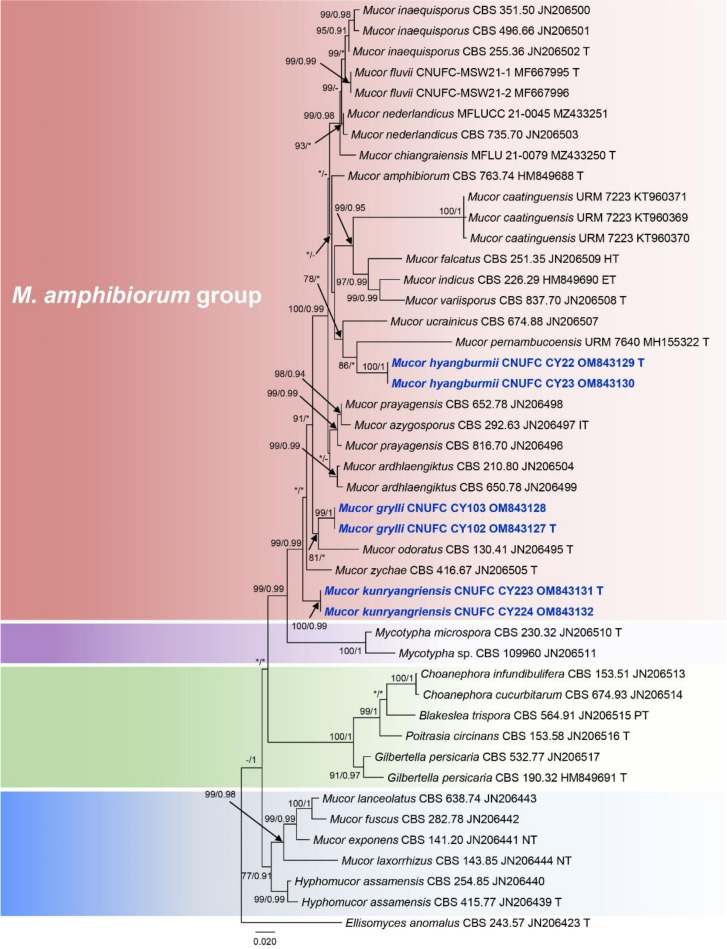
Phylogenetic tree constructed by maximum likelihood analysis of the LSU. The numbers above branches represent maximum likelihood bootstrap percentages (left) and Bayesian posterior probabilities (right). Bootstrap values ≥70% and Bayesian posterior probabilities ≥0.90 are shown. Bootstrap values lower than 0.90 and 70% are marked with “*”, and absent bootstrap values are marked with “-”. The bar indicates the number of substitutions per position. *Ellisomyces anomalus* CBS 243.57 was used as outgroup. Type, ex-neotype, ex-holotype, and ex-epitype strains are marked with T, NT, HT, and ET, respectively. Newly generated sequences are in blue bold font.

**Table 1 jof-08-00601-t001:** Taxa, collection numbers, and GenBank accession numbers used in this study.

Taxon Name	Strain Number	GenBank Accession Number		Host/Substrate	Country	Citation
		ITS	LSU			
*Blakeslea trispora*	CBS 564.91	-	JN206515	soil	China	[10]
*Choanephora cucurbitarum*	CBS 674.93	-	JN206514	n.a	China	[10]
*C*. *infundibulifera*	CBS 153.51	-	JN206513	n.a	n.a	[10]
*Ellisomyces anomalus*	CBS 243.57 (T)	JN205992	JN206423	dung of lizard	USA	[10]
*Gilbertella persicaria*	CBS 532.77	-	JN206517	dung of mouse	India	[10]
*G*. *persicaria*	CBS 190.32	-	HM849691	Prunus persica; fruit	USA	[10]
*Hyphomucor assamensis*	CBS 254.85	JN206212	JN206440	Burmannia	Malaysia	[10]
*H*. *assamensis*	CBS 415.77 (T)	JN206211	JN206439	n.a	India	[10]
*Mucor amphibiorum*	CBS 763.74 (T)	HM999957	HM849688	amphibian	Germany	[10]
*M*. *amphibiorum*	CBS 185.77	JN206170	-	diseased Dendrobates sp.	Central America	[10]
*M. azygosporus*	CBS 292.63 (T)	JN206187	JN206497	soil	USA	[10]
*M*. *ardhlaengiktus*	CBS 210.80 (ET)	JN206172	JN206504	garden soil	India	[10]
*M*. *ardhlaengiktus*	CBS 650.78	JN206174	JN206499	dung of lizard	India	[10]
*M*. *caatinguensis*	URM 7223	KT960375 KT960376 KT960377	KT960369 KT960370 KT960371	soil	Brazil	[26]
*M. chiangraiensis*	MFLUCC 21-0079 (T)	MZ433253	MZ433250	soil	Thailand	[5]
*M*. *exponens*	CBS 141.20 (NT)	JN206206	JN206441	n.a	Germany	[10]
*M*. *falcatus*	CBS 251.35 (HT)	JN206250	JN206509	honeycomb	Germany	[10]
*M*. *falcatus*	CBS 252.35	JN206249	-	dung of rabbit	Germany	[10]
*M*. *fluvii*	CNUFC-MSW21-2	MF667991	MF667996	freshwater	Korea	[11]
*M*. *fluvii*	CNUFC-MSW21-1 (T)	MF667992	MF667995	freshwater	Korea	[11]
*M*. *fuscus*	CBS 282.78	JN206201	JN206442	cheese	France	[10]
* **M.** * * **grylli** *	**CNUFC CY102 (T)**	**OM868230 (c1)** **OM868231 (c2)**	**OM843127**	* **Gryllus** * **sp.**	**Korea**	**This study**
* **M.** * * **grylli** *	**CNUFC CY103**	-	**OM843128**	* **Gryllus** * **sp.**	**Korea**	**This study**
* **M.** * * **hyangburmii** *	**CNUFC CY22 (T)**	**OM868232 (c1**) **OM868233 (c3)**	**OM843129**	* **Gryllus** * **sp.**	**Korea**	**This study**
* **M.** * * **hyangburmii** *	**CNUFC CY23**	-	**OM843130**	* **Gryllus** * **sp.**	**Korea**	**This study**
*M*. *inaequisporus*	CBS 255.36 (T)	JN206177	JN206502	Spondias mombin; fruit	Ghana	[10]
*M*. *inaequisporus*	CBS 496.66	JN206179	JN206501	Diospyros kaki; immature fruit	Japan	[10]
*M*. *inaequisporus*	CBS 351.50	JN206178	JN206500	Musa sapientum; fruit	Indonesia	[10]
*M*. *indicus*	CBS 226.29 (ET)	HM999956	HM849690	n.a	Switzerland	[10]
*M*. *indicus*	CBS 671.79	JN206183	-	n.a	Indonesia	[10]
*M*. *indicus*	CBS 120585	JN206180	-	human; muscle	India	[10]
* **M.** * * **kunryangriensis** *	**CNUFC CY223 (T)**	**OM868234 (c1)** **OM868235 (c2)** **OM868236 (c3)**	**OM843131**	* **Gryllus** * **sp.**	**Korea**	**This study**
* **M.** * * **kunryangriensis** *	**CNUFC CY224**	-	**OM843132**	* **Gryllus** * **sp.**	**Korea**	**This study**
*M*. *lanceolatus*	CBS 638.74	JN206205	JN206443	cheese	France	[10]
*M*. *laxorrhizus*	CBS 143.85 (NT)	JN206209	JN206444	lake mud	UK	[10]
*M*. *nederlandicus*	CBS 735.70	JN206176	JN206503	n.a	n.a	[10]
*M*. *nederlandicus*	MFLU 21-0078	MZ433254	MZ433251	soil	Thailand	[5]
*M*. *pernambucoensis*	URM 7640 (T)	MH155323	MH155322	soil	Brazil	[27]
*M*. *prayagensis*	CBS 652.78	JN206189 (c1) JN206190 (c3)	JN206498	dung of shrew	India	[10]
*M*. *prayagensis*	CBS 816.70 (T)	JN206188	JN206496	n.a	India	[10]
*M*. *odoratus*	CBS 130.41 (T)	JN206197	JN206495	laboratory air	Denmark	[10]
*M*. *odoratus*	CBS 201.71	JN206198	-	dung of horse	Netherlands	[10]
*M*. *odoratus*	CBS 120.71	JN206195	-	n.a	USA	[10]
*M*. *ucrainicus*	CBS 674.88	JN206192	JN206507	soil of litter layer	Germany	[10]
*M*. *ucrainicus*	CBS 221.71 (T)	JN206191	-	dung of mouse	Ukraine	[10]
*M*. *zychae*	CBS 416.67 (T)	JN206199	JN206505	manured soil	India	[10]
*M*. *variisporus*	CBS 837.70 (T)	JN206175	JN206508	n.a	India	[10]
*Mycotypha microspora*	CBS 230.32 (T)	-	JN206510	Citrus aurantium; peel, contaminant	Netherlands	[10]
*Mycotypha* sp.	CBS 109960	-	JN206511	human; pus of wound	Thailand	[10]
*Poitrasia circinans*	CBS 153.58 (T)		JN206516	soil	Trinidad and Tobago	[10]

Isolates and accession numbers determined in the current study are indicated in bold. CBS: Centraalbureau voor Schimmelcultures, Utrecht, The Netherlands; CNUFC: Chonnam National University Fungal Collection, Gwangju, Korea; MFLUCC: Mae Fah Luang University Culture Collection, Chiang Rai, Thailand; URM: Micoteca URM, Universidade Federal de Pernambuco, Recife, Brazil. Type, ex-neotype, ex-holotype, and ex-epitype strains are denoted by T, NT, HT, and ET, respectively.

## Data Availability

The sequencing data were submitted to GenBank.

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
