# Peer review of "Discovery of Three New Mucor Species Associated with Cricket Insects in Korea"

_jof, 2022, doi:10.3390/jof8060601_

Round 1
Reviewer 1 Report
The authors present three new species of the genus Mucor. This part of the fungal tree of life is exciting, and more taxonomic progress is clearly needed here. The manuscript is well-written and of an appropriate length.
Line 16 and 19. “columellae:” > “columellae, which are”
- “loci” is a genetic term that does not mean “genetic marker”. Thus, “loci” > “genetic markers”.
- By not writing “Mucorales” in italics, the authors violate the ICTF recommendation to write all fungal names in italics (https://imafungus.biomedcentral.com/articles/10.1186/s43008-020-00048-6).
- The authors are not consistent in their use of the Oxford comma. They use it on line 21 but not on, e.g., line 39. I thus propose: “group and” > “group, and” here. (comma also missing on line 45 and so on). Please resolve this in a consistent way throughout the manuscript.
- I think the reader deserves to know how the fungi were isolated. Please provide some more information here, rather than just referring to two other papers.
- It is customary to check newly generated sequences for basic quality (https://www.creamjournal.org/pdf/Cream_3_1_1.pdf) and to make a note of that in the manuscript. The present study does not do that. Does that mean that the newly generated sequences were not screened for quality? Is there, then, a risk that the new species are based on artefactual sequence data?
105-108. It is customary to partition MrBayes analysis of the ITS region in order not to average out information. I propose that the authors estimate three models for the ITS region (ITS1, 5.8S, and ITS2) and then do a partitioned MrBayes analysis. Also, please let it run for at least 5 million generations. If the resulting tree comes out the same (topology-wise) as the present three, then I would feel OK with the authors keeping the present tree in the study.
Why not also compute an ITS+LSU tree? Or were there too few specimens/cultures for which both ITS and LSU were available (from the same specimen/culture)?
- How was rooting/outgroup choice done?
- “in ITS” > “in the ITS”
125 and 127. “in LSU” > “in the LSU”
128-131. Please rewrite this sentence for clarity.
- ML is a quick and dirty method of phylogenetic inference when compared to MrBayes. Yet the ML trees rather than the Bayesian trees are shown. Why?
154, 188, and 220. “Gryllus” should be given in italics. Too bad that the authors didn’t sequence the cricket too, such that they would have been able to put a species name on it. Is there any hope for a morphological species identification of the crickets by some relevant cricket expert at the university of the authors?
179, 211, and 238. Species names should be given in italics.
- “Ph.D” > “Ph.D.”
- “within” > “within the”
244-246. Does this mean that Walther et al. knew about these tree new species and their phylogenetic affiliations but chose not to describe them? If not, please clarify this sentence.
- “in the” > “on an”
- The authors violate the MIAPA phylogenetic standard (https://pubmed.ncbi.nlm.nih.gov/16901231/) by withholding the multiple sequence alignments + phylogenetic trees. These should be released in TreeBase or Dryad (https://academic.oup.com/sysbio/article/69/6/1231/5820981 ; https://mycokeys.pensoft.net/articles.php?id=56691).
What can be said of the ecology of the new species? The word “ecology” is not even mentioned in the manuscript, contrary to the ICTF recommendations (https://imafungus.biomedcentral.com/articles/10.1186/s43008-021-00063-1 ; https://link.springer.com/article/10.1007/s11557-022-01796-y).
Reviewer 2 Report
Review on “ Discovery of Three New Mucor Species Associated with Cricket Insects in Korea” by
Thuong T. T. Nguyen and Hyang Burm Lee
The manuscript represents a very good taxonomic work introducing three novel Mucor species within the Amphibiorum spp. group. I do not have practically any objections towards this work only some minor notes bellow:
- Can authors please add the ITS similarities in % to the phylogenetic closest species (into discussion section) ?
- Please type all latin names with italics (through the text as well as the figure captions), this could be happening also by loading into journal format.
- Of information-description value would be to add the length of sporangiophores (or the high of colonies)
- After acceptation, register the fungi into MycoBank (add the MB number)
- I personally very much like the species epithet “ hyangburmii” devoting to one of the senior authors, for such a wonderful fungal taxon
- I do like also the information on the maximal temperatures; photomicrographs are really spectacular
-----end----
Author Response
Response to reviewer
Thank you so much for kindly giving comments and helpful revisions. I have carefully revised and corrected the manuscript. Attached please find the revised text in blue.
Here are answers to the comments and questions as follows:
The manuscript represents a very good taxonomic work introducing three novel Mucor species within the Amphibiorum spp. group. I do not have practically any objections towards this work only some minor notes bellow:
- Can authors please add the ITS similarities in % to the phylogenetic closest species (into discussion section) ?
Response: We would like to thank the reviewer. We added the information to the text. Please see revised manuscript.
- Please type all latin names with italics (through the text as well as the figure captions), this could be happening also by loading into journal format.
Response: We corrected it.
- Of information-description value would be to add the length of sporangiophores (or the high of colonies)
Response: According to the reviewer’s suggestions, we added the information to the text.
- After acceptation, register the fungi into MycoBank (add the MB number)
Response: We agree with the reviewer’s comment.
- I personally very much like the species epithet “ hyangburmii” devoting to one of the senior authors, for such a wonderful fungal taxon
- I do like also the information on the maximal temperatures; photomicrographs are really spectacular
We would like to thank the reviewer who read the manuscript carefully and gave kind and helpful comments.
With kind regards,
Hyang Burm Lee PhD
Dept. of Agricultural Biological Chemistry,
College of Agriculture and Life Sciences,
Chonnam National University, Gwangju 61186, South Korea
+82-062-530-2136 (TEL); +82-062-530-2139 (FAX); E-mail: hblee@jnu.ac.kr

Round 2
Reviewer 1 Report
I am, by and large, happy with the revised manuscript. However, if the authors keep refusing to make the underlying multiple sequence alignments available, then I will recommend the study for rejection (see below).
Line 13. Like I said, the ICTF recommendation is to write all fungal Latin names in italics. ”Mucorales” should thus be in italics here.
26. “described” > “was described”
26. “, and is” > “, and it is”
26. “Mucoraceae” should be given in italics.
42. “, and ” > “ and ”
51. “insect habitats” > “insects”. An “insect habitat” could mean “forest” or “meadow”, after all.
57. “bag” > “bags”
58. “from” > “from the”
59. “Bodies” > “The insect bodies”
81. “LROR” > “LR0R” (see https://mycokeys.pensoft.net/articles.php?id=4852&journal_name=mycokeys)
101. GenBank should be cited formally (see https://academic.oup.com/nar/article/50/D1/D161/6447240).
108. “, and ” > “ and ”
110. “three million” – but in their point-to-point reply, the authors said “five million” ?
110. If TreeBase is not working, then please submit the multiple sequence alignment to Dryad or as a supplementary file with the manuscript. Withholding the multiple sequence alignment is a violation of the MIAPA standard.
134. The word “basal” should not be used like that (https://resjournals.onlinelibrary.wiley.com/doi/10.1111/j.0307-6970.2004.00262.x). Please use the “sister clade” concept instead.
130, 136. “with a” > “with a moderate”
190. What does “Zygomycetes” means these days? I suggest “Mucoromycota” instead.
249. “Three new Mucor species belonging to the M. amphibiorum group [10] (Figures 1 and 2).” > “Three new Mucor species belonging to the M. amphibiorum group [10] are described (Figures 1 and 2).”
252. “with” > “with a”
254. “search” > “searches”
254 (and similarly for 268-269 and 286-287). I propose: “the ITS and LSU sequences of M. grylli were the most similar to”
261, 276, 277, 281, 299. “bigger” > “larger”
266. “are not” > “are not of”
281. “than that of” > “than those of the isolate”
303. “in the insect host” > “collected on insects”.
303. “three” > “the three”
314. “and is” > “and”
315. “discover” > “discover further”
